

# Horse phenotyping based on video image analysis of jumping performance for conservation breeding

Dorota Lewczuk[1] and Ewa Metera-Zarzycka[1,2]

[1] Institute of Genetics and Animal Breeding PAS, Jastrzebiec, Poland
[2] Bioekspert Sp. z o. o., Warszawa, Poland

## ABSTRACT

**Background**. Many horse breeds in the world are reserved as genetic resources; however, their characteristics seem to be insufficiently clarified, especially in terms of horse performance. Two jumping ability evaluation methods have been used to compare different types of performance breeds and on this basis their applicability for precision phenotyping has been determined.

**Methods**. Jumping data of 186 young Polish Warmblood stallions (27 with an endangered status) bred for sport and multipurpose use was collected during their performance tests organised under identical environmental conditions following the same guidelines. Jumping data consisted of objective measurements of free jumping parameters and the marks for jumping. Video recordings of 514 jumps (73 records for 27 stallions with an endangered status) were collected using a digital Panasonic AG-EZ 35 camera (25 fr/sec). Filming was recorded during a free jumping test in the line on a doublebarre obstacle (100–120 cm × 100 cm). Spatial and temporal variables of the jump were measured. The analysis of variance was performed (SAS, General Linear Model and Mixed procedures) using the statistical model, which included the random effect of the horse and fixed effects of the year of test, breeding status, height of jump and the successive number of the jump for objective kinematic data. The fixed effects of the year of test and breeding status were included in the model for subjective performance test data.

**Results**. Performance marks for free jumping were lower in the endangered group of stallions in the trainers' opinion ($p \leq 0.05$), while no statistically significant differences were found in the judges' opinions. Statistically significant differences in jumping variables were measured for the bascule points—the elevations of the withers and croup were higher in the endangered group ($p \leq 0.001$) and the take-off time was prolonged ($p \leq 0.05$), which explained the subjective evaluation.

**Discussion**. The use of objective evaluation methods provides important information for practice, as phenotypic differences between horses may be unclear in the subjective evaluation. The objective evaluation should be used to characterise the performance potential of different breeds, because the information from the evaluators might not be consistent. Such characteristics should be recorded at least for every new population.

Corresponding author
Dorota Lewczuk, d.lewczuk@ighz.pl

## INTRODUCTION

Horse genetic resources play an important role in horse breeding worldwide. Most horse breeds traditionally kept to preserve their historical, regional and local utility value act as gene pools in the case of a breeding crisis. An inbreeding depression can be a problem not only in small, closed populations, but also in large populations of closely related animals. Increased homozygosity in highly selected animals not only causes inbreeding depression (*Charlesworth & Willis, 2009*), but also fitness problems (*Leroy, 2014*). While it may not be an issue in current Warmblood horse breeding (*Borowska & Szwaczkowski, 2015*; *Pikuła et al., 2017*), several endangered subpopulations have been established as gene pools. Introduction of genetically unrelated horses within the same performance type would be the first step to follow in the case of inbreeding depression. Therefore, various horse breeds are preserved as national genetic resources (*Kompan et al., 2014*). These include animals of a local traditional performance type such as the Lusitano (*Vicente, Carolino & Gama, 2012*), old classical dressage horses such as the Menorca or Iberian (*Solé et al., 2013a*; *Solé et al., 2013b*; *Valera et al., 2013*), carriage horses e.g., the Lipizzaners (*Curik et al., 2003*) or Kladrubers (*Vostrá-Vydrová et al., 2016*) and others, such as e.g., the Finnhorse (*Sairanen et al., 2009*), Sorraia (*Kjöllerström, Gama & Oom, 2015*) or Friesian horses (*Ducro et al., 2006*). It seems necessary to monitor not only the genetic distance and genetic parameters of these populations, but also their usability and differences in performance. Such monitoring seems especially important for breeds kept as multipurpose genetic resources in relation to highly specialised sport populations (*Kompan et al., 2014*) such as the Tori horse from Estonia, the Napoletano from Italy, the Slovak Warmblood from Slovakia, the Gelder from the Netherlands or the Polish Małopolski and Wielkopolski subpopulations. New evaluation methods may improve the assessment of horse performance and consequently find applications in characterising endangered populations (*Kristjansson et al., 2013*; *Solé et al., 2013c*; *Kristjansson et al., 2016*). Studies mentioned above focused on gait characteristics, while the aim of the presented study is to characterise jumping performance based on video image analysis of horse breeds differing in terms of their endangered status and skills. Objectively measured jumping variables and subjective marks given by judges were used to test the hypothesis that the objective method will ensure more precision phenotyping of horses in terms of their jumping skills and will provide insight into jumping characteristics of endangered horses. Several attempts have been made in horse selection to monitor jumping characteristics of different breeds (*Lewczuk, 2008*; *Janczarek, 2011*); however, objective video image analysis (VIA) has never been used to monitor or compare endangered horse characteristics. The unique character of Polish tests gave such a possibility and facilitated a discussion of horse jumping skills. Both groups of the investigated horses are expected to be performance horses, the "non-endangered" population on a high sport competition level and the "endangered" on an overall riding and recreational level, thus differences between their jumping skills should be clearly visible. The phenotype description will also be compared using correlation analyses.

## MATERIAL & METHODS

Performance data were collected for 3-year old stallions, of which 159 were registered as sport breed Warmblood and 27 stallions of the endangered Wielkopolski and Małopolski subpopulations. All the horses were tested during the same 100-day performance tests at two training centres and the investigations were carried out over a period of three years. Horses were filmed in free jumping during the performance tests in order to measure jumping parameters and they were evaluated during their performance tests by the trainer, the judging committee and riders being specialists in their jumping discipline. According to the Polish law (3rd Ethical Local Commission Warsaw, Poland) observational research on practical procedures does not require any special approval. Free jumping was organized according to the scheme for performance tests of the Polish Horse Breeders Association—PHBA (http://www.pzhk.pl) on the final test day. A total of 514 jumps were recorded using Panasonic AG-EZ 35 (25 fr/sec) digital equipment, filming the third and last obstacle of a jumping line during the official performance test. The camera was standing in the same position 10 m from the horse pathway in the line of the obstacle centre. The filmed doublebarre obstacle was 100–120 cm high and 100 cm wide. The jumping line started with a ground pole, followed by two vertical obstacles of 50 cm and 65 cm in height. The distances between the obstacles were as follows: a pole on the ground 2.5 m before the first vertical, then 5.8–6.5 m before verticals and 5.8–7 m before the last filmed obstacle. The following jumping variables were measured on the frames of the films:

1. distances of take-off (the last full contact of both hind hooves before the airborne phase) and landing (the first full contact of both front hooves after the airborne phase);
2. distances of lifting for each limb (from the highest point of the pole to the lowest point of the hoof)
3. distances of elevation of the head, withers and croup (the highest points of these body parts on the "bascule" frame, where the withers are the highest body part in relation to the head and croup);
4. the angle of the head on the "bascule" frame (between the line of the nose and the line perpendicular to the ground);
5. the time of take-off and landing and the total time of the jump (the number of frames).

Spatial measurements were obtained using the manual program for image analysis developed by Cytowski (the Institute of Computer Science of the Polish Academy of Science) and temporal measurements by Motion DV Panasonic. The calibration for linear measurements was performed by measuring the distances between static marks on the ground on the horse jumping line.

Subjective judgment notes of the performance test results were obtained from the Polish Horse Breeders Association database. The performance test data consisted of jumping notes in a scale from 0 (very bad) to 10 points (excellent) for the following traits:

1. free jumping and jumping under the rider (as judged by the trainer and judges);
2. trainability (evaluated by the trainer);
3. rideability for jumping (evaluated by experienced riders hired for the test).

The obtained results were analysed statistically by the analysis of variance (Statistical Analysis Software; General Linear Mean and Mixed procedures). The statistical model for the objective data included the random effect of the horse and fixed effects of the status of breed (endangered/non-endangered), year of test, height of obstacle and the successive number of jump over the obstacle. The statistical model for the subjective data analysed separately included only fixed effects of the horse breed status (endangered/non-endangered) and the year of test. The following model was used for the objective data:

$$y_{ijklmn} = \alpha + B_i + Y_j + H_k + J_l + h_m + e_{ijklmn}$$

where:

$y_{ijklmn}$ - evaluation of the stallion,

$\alpha$ - adequate intercepts,

$B_i$ - fixed effect of ($i = 1,2$) for the endangered and non-endangered breed;

$Y_j$ - fixed effect of the year ($j = 1,2,3$) of the test;

$H_k$ –fixed effect of obstacle height ($k = 100,110, 120$);

$J_l$ –fixed effect of the successive number of the jump ($l = 1,2,3$);

$h_m$ –random effect of the horse ($m = 1,\ldots, 186$);

$e_{ijklmn}$ –random errors.

For the subjective data the model was as follows:

$$y_{ijkl} = \alpha + B_i + Y_j + e_{ijk}$$

where:

$y_{ijk}$ - evaluation of the stallion,

$\alpha$ - adequate intercepts,

$B_i$ - fixed effect of ($i = 1,2$) for endangered and non-endangered breed;

$Y_j$ - fixed effect of the year ($j = 1,2,3$) of the test;

$e_{ijkl}$ –random errors.

The interactions between main effects were tested and found not statistically significant. Differences between the levels of investigated effects were tested by multiple comparisons based on the all-pairs test between the least square means (LSM). For a detailed comparison of the data the results of the analysis of variance were supported by the partial correlations provided by the model (the MANOVA option in the Mixed procedure) for the relationships between notes and parameters, as well as those between jumping parameters for separate breeds.

## RESULTS

The characteristics of variables were normally distributed. The mean of all subjective notes was calculated individually, with the lowest of 5.5 and the highest score of 7.0 points and the standard deviation of 0.9–1.9. The take-off and landing distances were 266.2. two cm (SD 39.7) and 201.8. eight cm (SD 46.5), respectively. The height to which each limb was lifted ranged from 23.9 to 25.9. nine cm (SD 12-14.3). The mean elevation of the body for the measured bascule points was calculated in the same way to range between 122.9 and 132 cm (with SD 17-22). The mean head angle was 27.5 ° (SD 6.3). The mean duration of the jump was 16.6 frames (SD 2).

**Table 1 The least squares means (LSM) and standard errors (SE) for the effect of horse endangerment status on the objective jumping variables evaluated by video image analysis on stallions performance tests.**

| Traits | Status of endangerment LSM (SE) | |
|---|---|---|
| | Non-endangered (N = 159) | Endangered (N = 27) |
| Take off (cm) | 275.3 (13.4) | 279.2 (14.1) |
| Landing (cm) | 185.3 (15.3) | 182.2 (16.1) |
| Lifting of front right (cm) | 21.09 (3.7) | 20.1 (3.9) |
| Lifting of front left (cm) | 21.1 (3.7) | 19.4 (3.9) |
| Lifting of hind right (cm) | 23.0 (4.8) | 26.0 (5.0) |
| Lifting of hind left (cm) | 26.8 (4.8) | 28.0 (5.0) |
| Elevation of croup (cm) | 115.5 (5.0)[A] | 120.6 (5.4)[A] |
| Elevation of withers (cm) | 126.6 (5.4)[A] | 132.3 (5.7)[A] |
| Elevation of head (cm) | 124.4 (6.2) | 125.8 (6.6) |
| Head angle (°) | 28.6 (2.1) | 28.6 (2.2) |
| Take off time (frames) | 7.87 (0.37)[a] | 8.14 (0.39)[a] |
| Landing time (frames) | 7.91 (0.33) | 7.88 (0.34) |
| Total time (frames) | 16.78 (0.54) | 17.02 (0.56) |

**Notes.**
[A,a] Differences in rows statistically significant for $P \leq 0.001$ capitals. for $P \leq 0.05$ small letters.

Results of the statistical analysis are presented in Table 1 for objective measurements and in Table 2 for subjective performance test marks. Studied groups of horses differed statistically significantly in terms of their jumping style for the bascule points –the elevation of the withers and croup ($p \leq 0.001$) and for the time of take-off ($p \leq 0.05$). The group of horses from the endangered population elevated their bodies above the obstacle higher than the group of sport horses (above 4% in both variables) with the head position at the same height above the obstacle. The take-off time was longer for the endangered population (about 3%).

Statistically significant differences for the subjectively evaluated marks are observed for the following traits: trainability, jumping under the rider ($p \leq 0.001$) and free jumping ($p \leq 0.05$) evaluated based on the trainers' opinion. The non-endangered population received higher scores in free jumping (about 6%), while judges evaluated the same trait similarly. The notes for jumping skills in the performance test referred to the full 100-day performance period. Therefore, the trainers had a greater possibility to observe horses throughout the whole period of the test. Judges and riders visit the training centre once up to three times during the test. More information on jumping skills is received based on the measured objective jumping variables rather than objective judging, which also seems to vary in the amount of provided information.

Obtained correlations support these results as well. The correlations between measured parameters and marks for jumping traits received by both groups of horses are presented in Table 3. Among 13 traits only two were judged almost in the same way in both groups—it was lifting of both front limbs. The take-off distance was significantly correlated with most evaluated traits in the endangered horses. Landing was highly significantly correlated

**Table 2   The least squares means (LSM) and standard errors (SE) for the effect of horse endangerment status on subjective jumping evaluations on stallions performance tests.**

| Trait / evaluator (points) | Status of endangerment LSM (SE) | |
|---|---|---|
| | Non-endangered (N = 159) | Endangered (N = 27) |
| Trainability / trainer | 7.06 (0.08)[A] | 6.54 (0.18)[A] |
| Free jumping / trainer | 7.04 (0.07)[a] | 6.65 (0.17)[a] |
| Jumping with the rider / trainer | 6.74 (0.07)[A] | 6.15 (0.18)[A] |
| Free jumping / judges | 6.93 (0.07) | 7.13 (0.17) |
| Jumping with the rider / judges | 6.87 (0.07) | 6.87 (0.18) |
| Rideability jumping / riders | 6.00 (0.13) | 5.54 (0.32) |

Notes.
[A,a]Differences in rows statistically significant for $P \leq 0.001$ capitals. for $P \leq 0.05$ small letters.

with the free jumping evaluation score given by the judges (correlation coefficient 0.40) for the endangered group and only 0.09 for the other population. Correlations between free jumping and lifting of hind limbs were comparable for the scores given by the judges (approx. 0.2), but not for the trainers' notes. The correlation between the elevation of the croup and the note for jumping under a rider for the endangered group evaluated by the judges was twice as high as the respective correlations for the other horse group. The head angle and landing time received different scores in both groups as well. Most of the differences for these parameters are relatively small and seem random. These discrepancies might have been caused by the awareness of the catalogue information on individual horses and by personal preferences of the judges for specific horse lines.

In the comparison of relationships within jumping parameters (Table 4) most spatial parameters have lower values of correlations between jumping parameters for the endangered horses than for the non-endangered horses. That fact means that non-endangered horses are more elastic in jumping, as the parameters are not so dependent on each other. For the temporal parameters their relationships are not so obvious. The spatial parameters of jumps are time-correlated in both groups (to a greater extent for the endangered horses), whereas for the endangered group it was with the take-off time mainly and for the other population it was with the landing time. This may also be connected with the greater potential and elasticity of the jump, as it is known that basic parameters of the jump are dependent on the quality of the take-off.

## DISCUSSION

The horses from the endangered breed moved their trunk higher above the obstacles and either received lower marks for free jumping or did not differ in the scores based on the opinion of different evaluators. The higher elevation of the body is very often believed to be a sign of respect or even fear of obstacle. Horses are expected to jump economically –not too high above the obstacle; however, such a jumping style should be expected when sport horses are older and more experienced. The ability to recognise the height of an obstacle is considered to be an important horse skill (Kampman et al., 2012). Probably such preferences caused lower horse scores for free jumping in our study based on the trainers'

**Table 3** The correlations between measured parameters and results of evaluation for endangered and non-endangered groups of horses (statistically significant for $p < 0.05$ in bold).

| Jumping trait | Breed status | Performance traits evaluated by trainers -T. judges -J and riders -R | | | | | |
|---|---|---|---|---|---|---|---|
| | | Free jumping T | Free jumping J | Jumping under rider T | Jumping under rider J | Trainability T | Rideability T |
| Take off (cm) | endangered | 0.08 | **0.13** | **0.16** | **0.24** | **0.18** | **0.17** |
| | non-endangered | 0.01 | **0.15** | −0.02 | 0.13 | −0.06 | 0.01 |
| Landing (cm) | endangered | 0.05 | **0.40** | 0.15 | 0.16 | −0.03 | 0.19 |
| | non-endangered | 0.02 | **0.09** | −0.03 | 0.06 | −0.03 | −0.04 |
| Lifting of front right (cm) | endangered | **0.25** | **0.43** | **0.40** | **0.40** | **0.26** | **0.23** |
| | non-endangered | **0.29** | **0.30** | **0.23** | **0.25** | **0.25** | **0.19** |
| Lifting of front left (cm) | endangered | 0.23 | **0.33** | **0.22** | **0.30** | 0.09 | 0.23 |
| | non-endangered | **0.27** | **0.29** | **0.23** | **0.22** | **0.23** | **0.16** |
| Lifting of hind right (cm) | endangered | 0.07 | **0.35** | **0.27** | **0.38** | 0.11 | **0.21** |
| | non-endangered | **0.20** | **0.27** | **0.17** | **0.21** | **0.16** | **0.11** |
| Lifting of hind left (cm) | endangered | 0.08 | **0.34** | 0.18 | 0.32 | 0.02 | 0.19 |
| | non-endangered | **0.22** | **0.29** | **0.22** | **0.20** | **0.20** | **0.07** |
| Elevation of croup (cm) | endangered | 0.13 | 0.18 | 0.19 | **0.31** | 0.11 | 0.04 |
| | non-endangered | **0.10** | **0.21** | 0.06 | **0.13** | 0.05 | −0.02 |
| Elevation of withers (cm) | endangered | 0.11 | **0.21** | 0.18 | **0.28** | 0.11 | 0.07 |
| | non-endangered | **0.10** | **0.20** | 0.04 | **0.12** | 0.02 | −0.03 |
| Elevation of head (cm) | endangered | −0.15 | −0.12 | −0.09 | 0.07 | 0.04 | −0.10 |
| | non-endangered | **−0.08** | −0.03 | −0.14 | −0.08 | **−0.09** | **−0.17** |
| Head angle (°) | endangered | −0.06 | **−0.24** | **−0.24** | −0.10 | −0.04 | **−0.34** |
| | non-endangered | **−0.18** | **−0.14** | **−0.21** | **−0.13** | **−0.19** | **−0.23** |
| Take off time (frames) | endangered | **0.20** | −0.17 | 0.19 | **0.23** | **0.47** | 0.01 |
| | non-endangered | **0.11** | **0.13** | **0.10** | **0.11** | **0.09** | **0.09** |
| Landing time (frames) | endangered | 0.06 | **0.16** | 0.18 | **0.30** | 0.04 | 0.01 |
| | non-endangered | **0.29** | **0.32** | **0.21** | **0.22** | **0.15** | **0.16** |
| Total time (frames) | endangered | 0.17 | **0.22** | **0.25** | **0.35** | **0.33** | −0.01 |
| | non-endangered | **0.25** | **0.29** | **0.19** | **0.21** | **0.14** | **0.16** |

opinion (6.65 points in the endangered group vs. 7.04 in the non-endangered group). The higher elevation of the body (about 4%) of the endangered horses with their limbs being lifted to the same height as in the non-endangered horses resulted in the endangered horses considered to be jumping higher because of insufficient limb elasticity. That supported the trainers' lower opinion (0.4 points) on jumping skills of the endangered horses. Their opinion may have also been affected by the requirements of the most important jumping competitions that are conducted "against the clock". The higher trunk elevation takes more time than a lower jump, so horses jumping higher need more time to clear the course and may lose their competitions in the future. Such an attitude to jumping assessment complied with the KWPN guide for evaluating the quality of jumping technique. Both in the KWPN and in the PHBA guidelines the speed of take-off is considered a positive characteristic (*Kampman et al., 2012*).

Lewczuk and Metera-Zarzycka (2019), *PeerJ*, DOI 10.7717/peerj.7450

**Table 4** The correlations between measured parameters for endangered and non-endangered groups of horses (statistically significant for $p < 0.05$ in bold).

| Jumping trait | Breed status | Take off | Landing | Lifting of front right | Lifting of front left | Lifting of hind right | Lifting of hind left | Elevation of croup | Elevation of withers | Elevation of head | Head angle | Take off time | Landing time | Total time |
|---|---|---|---|---|---|---|---|---|---|---|---|---|---|---|
| Take off (cm) | endangered | X | **0.32** | **0.48** | **0.63** | **0.58** | **0.52** | **0.62** | **0.63** | **0.45** | −.26 | **0.48** | 0.09 | **0.38** |
| | non-endangered | X | 0.13 | 0.14 | 0.22 | −.08 | 0.07 | **0.54** | **0.53** | **0.37** | **0.20** | **0.33** | 0.14 | **0.32** |
| Landing (cm) | endangered | **0.32** | X | **0.47** | **0.48** | **0.55** | **0.58** | **0.59** | **0.58** | **0.30** | −.13 | **0.23** | 0.13 | **0.25** |
| | non-endangered | 0.13 | X | **0.15** | **0.15** | **0.11** | **0.16** | **0.38** | **0.43** | **0.35** | −.04 | −.06 | **0.12** | **0.13** |
| Lifting of front right (cm) | endangered | **0.48** | **0.47** | X | **0.85** | **0.77** | **0.77** | **0.72** | **0.70** | **0.33** | −.17 | **0.44** | **0.26** | **0.46** |
| | non-endangered | **0.14** | **0.15** | X | **0.66** | **0.25** | **0.28** | **0.48** | **0.48** | **0.33** | −.15 | **0.31** | **0.34** | **0.42** |
| Lifting of front left (cm) | endangered | **0.63** | **0.48** | **0.85** | X | **0.76** | **0.76** | **0.77** | **0.70** | **0.33** | −.17 | **0.38** | **0.26** | **0.34** |
| | non-endangered | **0.22** | **0.15** | **0.66** | X | **0.30** | **0.29** | **0.54** | **0.54** | **0.41** | −.13 | **0.31** | **0.36** | **0.43** |
| Lifting of hind right (cm) | endangered | **0.58** | **0.55** | **0.77** | **0.76** | X | **0.96** | **0.76** | **0.77** | **0.37** | −.19 | **0.39** | **0.43** | **0.54** |
| | non-endangered | −.08 | **0.11** | **0.25** | **0.30** | X | **0.91** | **0.30** | **0.25** | 0.03 | −.19 | 0.03 | **0.40** | **0.26** |
| Lifting of hind left (cm) | endangered | **0.52** | **0.58** | **0.77** | **0.76** | **0.96** | X | **0.77** | **0.77** | **0.34** | −.13 | **0.32** | **0.37** | **0.46** |
| | non-endangered | 0.07 | **0.16** | **0.28** | **0.29** | **0.91** | X | **0.37** | **0.35** | 0.08 | −.20 | 0.07 | **0.41** | **0.30** |
| Elevation of croup (cm) | endangered | **0.62** | **0.59** | **0.72** | **0.77** | **0.76** | **0.77** | X | **0.97** | **0.65** | −.01 | **0.54** | **0.37** | **0.60** |
| | non-endangered | **0.54** | **0.38** | **0.48** | **0.54** | **0.30** | **0.37** | X | **0.96** | **0.72** | −.05 | **0.32** | **0.26** | **0.38** |
| Elevation of withers (cm) | endangered | **0.63** | **0.58** | **0.70** | **0.70** | **0.77** | **0.77** | **0.97** | X | **0.63** | −.12 | **0.48** | **0.43** | **0.60** |
| | non-endangered | **0.53** | **0.43** | **0.48** | **0.54** | **0.25** | **0.35** | **0.96** | X | **0.70** | −.09 | **0.30** | **0.30** | **0.39** |
| Elevation of head (cm) | endangered | **0.45** | **0.30** | **0.33** | **0.39** | **0.37** | **0.34** | **0.65** | **0.63** | X | **0.34** | **0.46** | 0.10 | **0.39** |
| | non-endangered | **0.38** | **0.35** | **0.33** | **0.41** | 0.03 | 0.08 | **0.72** | **0.70** | X | **0.25** | **0.17** | 0.09 | **0.17** |
| Head angle (°) | endangered | −.26 | −.13 | −.17 | −.17 | −.19 | −.13 | 0.01 | −.12 | **0.34** | X | 0.06 | −.19 | −.09 |
| | non-endangered | **0.20** | −.04 | **−.15** | **−.13** | **−.19** | **−.20** | −.05 | −.09 | **0.25** | X | −.01 | **−.19** | **−.13** |
| Take off time (frames) | endangered | **0.48** | **0.23** | **0.44** | **0.38** | **0.39** | **0.32** | **0.54** | **0.48** | **0.48** | 0.06 | X | 0.13 | **0.74** |
| | non-endangered | **0.33** | −.06 | **0.31** | **0.31** | −.03 | −.07 | **0.32** | **0.30** | **0.17** | −.01 | X | **0.19** | **0.80** |
| Landing time (frames) | endangered | 0.09 | 0.13 | **0.26** | **0.26** | **0.43** | **0.37** | **0.37** | **0.43** | 0.10 | −.19 | 0.13 | X | **0.76** |
| | non-endangered | **0.14** | **0.12** | **0.34** | **0.36** | **0.40** | **0.41** | **0.26** | **0.30** | 0.09 | **−.19** | **0.19** | X | **0.73** |
| Total time (frames) | endangered | **0.38** | **0.25** | **0.46** | **0.36** | **0.54** | **0.46** | **0.60** | **0.60** | **0.39** | −.09 | **0.74** | **0.76** | X |
| | non-endangered | **0.32** | **0.13** | **0.42** | **0.43** | **0.26** | **0.30** | **0.38** | **0.39** | **0.17** | **−.13** | **0.80** | **0.73** | X |

In turn, the ability to jump higher and the lightness of jump ("ease" and "willingness") were the traits included in the judges' evaluation system as well. It is likely that a higher elevation and a better curve of the trunk (with the head being in the same position) observed in the endangered group of horses were treated as superior jumping abilities. Thus the marks given by the judges to horses jumping in that manner were slightly higher in the analysed performance results, while not being statistically significant (7.13 points for the endangered group vs. 6.93 for the non-endangered group).

The differences in scores for jumping under the rider in the trainers' opinion may be associated with the fact that the Polish endangered Warmblood horses are more closely related to Arabian horses, so they tend to be more temperamental and as such more difficult for riders to handle. In the presented study the endangered group was evaluated higher in terms of temperament - 7.97 points (SD 0.16) as compared with 7.65 (SD 0.06) for the other group (data not presented in the tables, the difference statistically non-significant).

Differences in the opinions on jumping skills may also be affected by the character of visual perception. The human perception is a complex process involving integration of the first and second order motion signals (e.g., luminance, contrast) from various parts of the retinal image carried by separate processing pathways. The "global motion" picture is produced from individual visual fields of neurons (*Burr & Santoro, 2001*). Especially fast motion may be connected with some difficulty in the correct assessment of the 'signal' and 'noise' required to determine motion. The human perception is connected with different phenomena. For example, if the special time period between separate image stimuli is used, the eyes provide the perception of motion; similarly, a fast succession of still images of a ball within a proper distance gives the illusion of a moving ball (*Steinman, Pizlo & Pizlo, 2000*). It seems possible that expectations of the judges based on the knowledge of the horse's pedigree as well as their individual preferences based on conformation or earlier experiences may determine their perception as well. In our study the trainers gave their notes within a wider scale (2–9 points) than the judges (4–10 points), which was probably because of their more extensive experience with the tested horses.

Linear evaluation systems commonly used in horse breeding seem less dependent on such influences. Probably that is the reason why they have been developed and implemented (*Koenen, Van Veldhuizen & Brascamp, 1995*; *Rustin et al., 2009*; *Duensing, Stock & Krieter, 2014*). Even if they do not provide the same level of accuracy as VIA methods, they may be considered an optimum tools (*Duensing, Stock & Krieter, 2014*). Characterisation of traits in the descriptive scale seems to be a better option for horse evaluation in comparison with the subjective evaluation system. Unfortunately, similarly as the point score system linear scoring is also based on the judge's experience, rather than measured values. Some further studies on the objectivity of linear scoring have been conducted (*Doucet, 2007*). Additionally, three-dimensional methods have been shown to be more accurate than two-dimensional ones, since they are not restricted to the calibration area (*Weller et al., 2006*). Nevertheless, they still seem of limited use under field conditions.

Every system has its limitations. The subjective definition of a trait does not give the same information for everybody, as judges differ in their opinions (*Lewczuk, 2013*). On the other hand, a detailed linear description of a trait may cause the loss of important information
not specifically named and recorded earlier. Both evaluation systems (linear scoring vs. traditional point system) may be influenced by different personal experiences of judges. In the linear system all traits are described in relation to the population mean being the centre of the scale (e.g., short vs. long legs; square vs. rectangular body frames), while in the point evaluation scale (from 1 - very bad to 10 - excellent) the note is referred to the ideal image as perceived by the judge. In both cases the evaluation depends on the experience of evaluators. Their 'population mean' as well as 'the concept of a perfect horse' depend on the experience and may potentially affect the actual comparisons between countries or breeds. Another complexity is related with the evolution of prospects in animal breeding. Traits may be changed in the ideal standards in response to the selection processes and as a consequence, in the population means. An example may be provided by the horse production area. Height at the withers is just such a characteristic. Straight lines between obstacles in show jumping courses with high obstacles used until the 1960's (*Gego, 2006*) favoured selection for horses that were high at the withers to enlarge the scope of jumps. It changed later as the maximal jumping scope was not the main difficulty in show jumping competitions. New trends in jumping course design changed conformation preferences in breeding of jumping horses. Difficult lines to follow between obstacles, curves of the horse's path in jumping courses created new demand for winner horses that had to be elastic, smaller and easily rideable. This characteristic was not correlated with the large body frames of horses bred earlier. Horses studied in our research did not differ in terms of their body frames (height at the withers - 166 cm for the non-endangered vs. 164 cm for the endangered group with SD of 2.31 and 2.88, respectively). Changes in the 'ideal' conformation influence the evaluation in selection as well. Thus none of these systems may guarantee an absolutely reliable comparison of different horse populations, especially over long time periods. From this point of view "non-idealised" evaluation systems based on measurements and independent of the observer's experience should be used to characterise endangered jumping breeds. Correlations between jumping parameters obtained in our study also underline the need for more precisely defined horse characteristics, as they show differences in the horse jumping technique not reflected in the judge notes. The level of the correlation coefficient presented in the study is comparable with the results for Polish warmbloods calculated for all breeds (*Janczarek, Stachurska & Wilk, 2013*); however, a detailed comparison is not possible as different types of parameters were studied (angular vs. linear).

Detailed and precise methods, although time-consuming, are employed in current horse breeding research to find new genetic mutations responsible for specific traits of the endangered horse population (*Kristjansson et al., 2016*). However, the genomic tool may be only as detailed and accurate as the phenotypic data used for the primary association analysis. The discrepancy in the results of genome mapping studies may reflect the variability in phenotypic criteria (*McCoy et al., 2016*). This provides a promising future for precision phenotyping. Using subjective marks, e.g., free jumping studied in our paper, may result in confusing results when traits are denoted by ambiguous terms. The functional nature of the breeding goal is always the main aim for horse breeding, including also endangered breeds (*Weber, 2005*). Monitoring of horses constituting the dressage genetic resource pool

facilitates introduction of particular sires into the programme (*Valera et al., 2013*). The same should apply to endangered horse populations used in jumping.

## CONCLUSIONS

Objective evaluation methods provide important additional information in horse practice and provide more insight than is revealed by experts. In our study the differences in free jumping scores between horses differing in their endangerment status when judged subjectively by experienced evaluators were evidenced and clarified based on the jumping variables measured using the video image analysis. Methods based on objective measurements provide precise characteristics for each breed and seem particularly important for the present day precise livestock strategies. Detailed characteristics of jumping performance for endangered horse breeds should be provided.

### Funding
This work was supported by statute project S.V.1 and NCN grant NN 311 33 46 36. The funders had no role in study design, data collection and analysis, decision to publish, or preparation of the manuscript.

### Grant Disclosures
The following grant information was disclosed by the authors:
statute project S.V.1.
NCN grant: NN 311 33 46 36.

### Competing Interests
The authors declared there are no competing interests exist.

### Author Contributions
- Dorota Lewczuk conceived and designed the experiments, performed the experiments, contributed reagents/materials/analysis tools, prepared figures and/or tables, authored or reviewed drafts of the paper, approved the final draft.
- Ewa Metera-Zarzycka conceived and designed the experiments, analyzed the data, contributed reagents/materials/analysis tools, prepared figures and/or tables, authored or reviewed drafts of the paper, approved the final draft.

### Animal Ethics
The following information was supplied relating to ethical approvals (i.e., approving body and any reference numbers):

According to the decision of the 3rd Ethical Local Commision (Warsaw, Poland) observational research on practical procedures do not need any approval.

## Ethics

The following information was supplied relating to ethical approvals (i.e., approving body and any reference numbers):

3rd Warsaw Ethical Commission

## Data Availability

The database may not be public according to the protection of the human rights and privacy.

All horses that attend the research belonged to the private people and information included in the datasets (even anonymous) may be used against the owners and the economic value of their horses may depend on the information included in the database.

Sharing large datasets or databases that contain information about human subjects presents a special challenge because of the requirement to protect the rights and privacy of people who participate in research studies.

The data were obtained on the basis of the agreement on the breeding value estimation between Institute of Genetics and Animal Breeding PAS and Polish Horse Breeders Association (dated 2002) which restricts the users from sharing the data on the grounds of privacy or safety. Please contact the corresponding author with any questions or interest in the data.

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
