# Peer review of "Horse phenotyping based on video image analysis of jumping performance for conservation breeding"

_PeerJ, doi:10.7717/peerj.7450_

## Round 0.1 · original submission · Major Revisions

First and foremost, it is absolutely essential for you to work with someone highly proficient at English language editing to thoroughly rewrite your manuscript from beginning to end. As it reads, I am unclear as to exactly what you did, and even more importantly why you did it. For example, consider this phrase from the abstract: "Precise phenotyping is the main condition for further precise genotyping so important for nowadays breeding." I do not know what you mean by this. There are many places throughout the manuscript that are similarly unclear. You need to make a clear case for phenotyping instead of doing genotyping. And you should respond in detail to all the comments from both reviewers.

Reviewer 1 ·

Basic reporting

The topic of the paper is important due to the problems connected with conducting preventive breeding of horses in different countries. One of such problems is the opinion on horses from such breeding that due to the stabilizing selection of their utility value is low. The situation may be harmful for those horses during subjective assessment of judges which subconsiously assess the individuals covered with protection assessed by assumption as worse. As it is commonly known, it is very often inconsistent with the real situation.
1. Generally the work may be published in Peer J, however the major revision will be necessary. First of all, the hypothesis in my opinion is wrongly formulated. The Authors should assume that horses are assessed worse despite their jump parameters are comparable to the undangered horses. The purpose of the paper is to compare the results of measurable and immeasurable assessment of two race groups, and then selection of the incoherent assessments with the jump parameters. Maybe in the future the assessments should not be taken into account and they should be replaced with others.

Experimental design

A description of the experiment is proper, however the results concerning average and SD contained in statistical methods should be transferred to the result chapter. Please provide the distance from the obstacle filmed. Did the filing take place the same day as the assessments of judges? It is not clear for me.

Validity of the findings

It is necessary to reedit the final arrangements for adjusting to the purpose of the paper proposed by me.

Additional comments

Please reedit the text with reference to the new purpose of the paper. The discrepancies between the subjective and objective assessment have already been proved long time ago. However, for different reasons the subjective assessment may not not be deviated. Therefore it is important to prove which assessments and given by whom are wrong and why. The discussion should go in this way. In the present form the discussion is too long. Some fragments thereof are unnecessary, e.g. in for which purpose the authors consider reasons for subjectivism of the judge assessments. This is not the topic of the paper. The subjectivism does not refer only the assessment of the endangered horses. Besides I propose to calculate the correlation between the measurable and immeasurable features, I think that in this case the reply to the thesis of the paper will be more specific. Besides, think whether these discrepancies between the assessments and parameters are not reasonable. E.g. in case of the endangered horses the beam was higher but the legs lower above the obstacle. Maybe the fact caused such assessments and not others and do not cover separately particular features of jump. The summary as a result should be reedited.

Reviewer 2 ·

Basic reporting

The English in this article requires significant improvement to be suitable for publication. This reviewer is a native English speaker who found themselves reading the same sentence three to four times just to try and make sense of it. I can only imagine that for someone whose first language is not English, this text could be very difficult to understand. Some examples from only the abstract are below:

Line 21 – ‘…their characteristics seem not enough recognised’ is not proper English, and could be changed to ‘their characteristics are not well recognised’

Line 21-23 – ‘Precise phenotyping is the main requirement for further precise establishing of association with their genotypes so important for nowadays breeding.’ I believe what the authors are trying to say is that modern horse breeders require a precise establishment of the genotype-phenotype relationship in horses. Furthermore, that current approaches for phenotyping of horses have fallen behind the approaches for genotyping. Whether that be in terms of accuracy, throughput or both, is not stated. However, this summary is not clear from the quoted sentence and is only my best guess.

Line 23 – ‘Therefore a data’. Data is the plural of datum. This should read ‘Therefore data of Polish…’.

Line 24 – ‘…used to compare two jumping availability evaluation methods’ is technically correct but is much easier to read as ‘…used to compare two methods for evaluating jumping ability’.

There are far too many places throughout the text where English needs to be improved so I won’t give detailed feedback on all of them. Instead I’ll highlight cases where the mistakes in English could be misleading rather than just inaccurate.

Line 32 – ‘Linear and temporal variables of the jump were measured’. Do you mean ‘spatial and temporal’?

Line 33 – Nowhere in the text is it stated that GLM is short for ‘Generalized Linear Model’ or SAS is short for ‘Statistical Analysis Software’. These abbreviations must be defined the first time they are used.

Line 41 – The text appears to have decreased in font size half way through this line?

Line 72-74 – This sentence does not make sense to me. In the beginning I guessed it meant that variations in horse genomics could cause variations in the horse phenotype, specifically in the phenotypic traits used for genomics analyses. But ‘even if the criteria are named equally’ is confusing.

Line 78 – My background is not in horses or horse phenotyping so this question may have an obvious answer, but why is endangerment status used as the independent variable here? As I understand it, endangerment refers to how close a species is to extinction? You have later started that all horses are expected to be performance horses, so it is not clear to me how the level of endangerment would have a causal relationship with jumping variables? If this is an obvious misunderstanding on my part that will be understood by the majority of readers then please excuse my ignorance on the matter, but otherwise perhaps make the premise clearer in the text.

Line 121 – ‘All data based on their characteristic were checked’, what does this mean? You checked that they were normally distributed? Or something else?

Line 121 – ‘The means of subjective data were between 5.5-7.0 points’, is this for all three (free jumping, trainability and rideability) variables? The mean of each variable was calculated individually and the lowest one is 5.5 and the highest 7.0? This is not clear.

Line 124 – ‘mean elevation of the body above the obstacle during the bascule frame was 122.9-132 cm’, how can mean elevation have a range of 9.1 cm? Wouldn’t the mean of a set of elevations be a scalar value?

Line 127 – ‘The obtained results were analysed statistically by the variance analysis’. I’m not familiar with ‘variance analysis’ as a general term or technique, and a quick google search only reveals applications in finance and accounting? After this you reference a statistical model but give no real information about the model. In brackets there is GLM and ‘Mixed procedures’, but this is insufficient detail. An equation with labelled variables may be helpful. Furthermore the Generalized Linear Model and the General Linear Model are two different models. Without an expansion on the abbreviation or an explanation of the model used, it’s impossible to tell which is employed.

Line 133-134 – ‘Differences between levels of investigated effects were tested by the multiple comparison based on all pair ways test between the least squares means’. It is not clear what is meant here. Perhaps this is because the statistical model was not made clear.

Results and discussion sections – Confusion here may again stem from a lack of clear English and a lack of detail in the statistical model, but it appears that all results are based around the differences in measurement between endangered and non-endangered horses. If this is the case, why did this require a statistical model? With sufficient data a p-value can be obtained using the data alone. It also seems that there is a lot more room for interpreting these specific results, whether it be in the results section or the discussion section. For example, the trainers marked the non-endangered population significantly higher when it came to free jumping. Whereas the objective measurements showed that the endangered population elevated their body significantly higher. The discussion section explains why this is the case, citing time spent in the air in ‘against the clock’ competitions and that horses are expected to jump economically. This argument, and its counter-argument take up a large portion of the discussion section but do not cite numbers from your own results once. In fact data or results appear only once in the discussion section, when there is opportunity for them to be used many times. Another example is the issue of subjectivity, where the authors argue that judges’ or trainers’ individual experiences will affect their marking. This would have been a good opportunity to reference or compare the standard deviation of judges’ and trainers’ marks for the different variables.

Once again forgive me if I have misunderstood the motivation or analysis of the work, but could the authors also perhaps have tried to fit a linear model between judge’s marks for free jumping and the objective variables associated with jumping? With the hypothesis that larger values in objective jumping variables such as distance of elevation and total jump time would have an inverse relationship with subjective marks? This doesn’t seem to have been done yet as Line 131 states that the statistical model for subjective data was analysed separately.

Experimental design

The design of the experiment is sound. The research question is well defined, relevant and meaningful. The emergence of automated and quantitative methods for phenotyping, to match the acceleration of genotyping techniques is present in many different fields. Using a video camera to measure different horse phenotypic traits and compare them to manual measurements will surely be a relevant study.

As described in the basic reporting section, the methods have not been described with sufficient detail and information to be replicated.

In terms of a rigorous investigation, for the subjective variables, the marks from the judges, trainers and riders seem to have been collected in an appropriate way. For the objective variables, data collection is also fine as long as the location and orientation of the camera is the same during all acquisitions. As the authors are recording two-dimensional variables from a three-dimensional scene perspective will play a part. For instance, if the camera is directly perpendicular to the jump, measuring height in centimetres using a linear scale placed at the same distance as the horse from the camera will be accurate. However pixels at different depths from the camera will have different real-world dimensions (cm’s) due to perspective, so the variable ‘lifting of every limb’ will not be accurate to the same “cm’s”. This is not an issue for this study if all horses are jumping from the same position and the camera does not move, as the distances used in the objective measurements will all have the same relative error. If the camera was set up at a new location, with a different distance to the jump and orientation, some error would be introduced.

Validity of the findings

I think that the findings and conclusions could have been much more clearly stated. The authors failed to use their own results to comprehensively show that their approach is necessary. There are two tables showing the difference in variable values between endangered and non-endangered horses for the subjective and objective variables, and they are shown independently. No real effort was made to show a correlation between the two. And, that analysis would not even require the distinction between endangered and non-endangered, but rather measurements at an individual horse measurement. Is the point of the study not to show that the objective measurements either complement the subjective ones, or reveal something that the subjective ones could not? Whether the data shows this, or it does not show it, is not discussed in depth.

Additional comments

If I understand it correctly I think the premise of the study is fine. But the English and presentation of results are so poor that I cannot tell if I understand it correctly.

I don't work in image-based phenotyping for horses but rather for plants. For us the goal is to use image analysis to either a) record large amounts of measurements that would agree with an expert automatically or b) uncover information that the experts could not. I think the premise of this paper is the same. But the results are so sparse, and so sparsely discussed and interpreted, I cannot tell if there is agreement between the subjective and objective measurements, or whether the objective measurements have found something new or they are more robust. This should be made more clear throughout the paper, in the introduction, results, discussion and conclusion.

---

## Round 0.2 · Major Revisions

I would like to reiterate an important point made by one of the reviewers: that it will be necessary for you to have this manuscript read and edited by a native-English writer before resubmitting it. There are too many instances where the language, sentence structure, and grammar are confusing to include in a single review. Although this will slow down your revision time, it is essential that you do this.

In addition, your statistical models only focus on main effects. Readers (including me) will be wondering why you did not consider the potential for interactions between the main effects. Please address this point in your revision, and if necessary perform additional analyses.

---

## Round 0.3 · accepted · Accept

Thank you for clarifying the statistical methods used in model selection and for improving the English grammar.